# Clinical and Demographic Features of Paracoccidioidomycosis in Argentina: A Multicenter Study Analysis of 466 Cases

**DOI:** 10.3390/jof9040482

**Published:** 2023-04-17

**Authors:** Gustavo Giusiano, Fernanda Tracogna, Gabriela Santiso, Florencia Rojas, Fernando Messina, Vanesa Sosa, Yone Chacón, Maria de los Ángeles Sosa, Javier Mussin, María Emilia Cattana, Andrea Vazquez, Patricia Formosa, Norma Fernández, Milagros Piedrabuena, Ruth Valdez, Florencia Davalos, Mariana Fernández, Alejandra Acuña, Alejandra Aguilera, Liliana Guelfand, Javier Afeltra, Guillermo Garcia Effron, Gladys Posse, Susana Amigot, Julian Serrano, Otilia Sellares, Christian Álvarez, Gloria Pineda, Susana Carnovale, Laura Zalazar, Cristina Canteros

**Affiliations:** 1Departamento Micología, Instituto de Medicina Regional, Universidad Nacional de Nordeste, CONICET, Resistencia 3500, Argentina; 2Hospital Pediátrico Juan Pablo II, Corrientes 3400, Argentina; 3Hospital Julio C. Perrando, Resistencia 1100, Argentina; 4Unidad de Micología, Hospital de Enfermedades Infecciosas F. J. Muñiz, Uspallata, Buenos Aires 2272, Argentina; 5Servicio de Micología, Hospital Dr. Ramon Madariaga, Av. Marconi 3736, Posadas N3300, Argentina; 6Hospital Señor del Milagro, Salta 4400, Argentina; 7Laboratorio Central de Redes y Programas, Facultad de Ciencias Exactas y Naturales y Agrimensura, Instituto de Medicina Regional, Universidad Nacional del Nordeste, Placido Martínez, Corrientes 1044, Argentina; 8Servicio de Microbiología, Hospital 4 de Junio Ramón Carrillo, Roque Sáenz Peña, Av. Malvinas Argentinas 1350, Sáenz Peña H3700, Argentina; 9Hospital de Alta Complejidad Pte. J. D. Perón, Av. Pantaleón Gómez & Av. Dr. Nestor Kirchner, Formosa 3600, Argentina; 10Laboratorio de Micología, Hospital de Clínicas, José de San Martin, Buenos Aires 2351, Argentina; 11Laboratorio de Microbiología, Hospital San Martín, Pres. Juan Domingo Perón 450, Paraná 3100, Argentina; 12Servicio de Microbiología, Hospital San Bernardo, Av. José Tobias 69, Salta 4400, Argentina; 13Hospital J. D. Perón, Tartagal, Salta 4560, Argentina; 14Hospital Materno-Infantil, Salta 1301, Argentina; 15Sección Microbiología, Hospital General de Agudos Dr. Juan A. Fernández, Buenos Aires 1425, Argentina; 16Unidad de Parasitología y Micología, Hospital General de Agudos José María Ramos Mejía, Ciudad Autónoma de Buenos Aires 1221, Argentina; 17Laboratorio de Micología y Diagnóstico Molecular, Cátedra de Parasitología y Micología, Facultad de Bioquímica y Ciencias Biológicas, Universidad Nacional del Litoral, CONICET, Santa Fe 2750, Argentina; 18Hospital Nacional Prof. Dr. A. Posadas, Buenos Aires 1684, Argentina; 19CEMAR Microbiología, Dir. Bioquímica, Secretaría de Salud Pública, Rosario 2020, Argentina; 20Sección Micología, Hospital Independencia, Av. Belgrano Nte. 660, Santiago del Estero 4200, Argentina; 21Hospital Central, Reconquista, S3560 Reconquista, Argentina; 22División Micología—Laboratorio de Salud Pública de Tucumán, Tucumán 4000, Argentina; 23Hospital Universitario Austral, Pilar, Buenos Aires 1500, Argentina; 24Hospital de Pediatría S.A.M.I.C. Prof. Juan P. Garraham, Pichincha 1890, Buenos Aires 1245, Argentina; 25Facultad de Humanidades, Universidad Nacional del Nordeste, Las Heras 727, Resistencia H3500COI, Argentina; 26Departamento de Micología, INEI-ANLIS “Dr. Carlos G. Malbrán”, Buenos Aires 1281, Argentina

**Keywords:** paracoccidioidomycosis, epidemiology, demographic, clinical, diagnosis, Argentina

## Abstract

Information on paracoccidioidomycosis (PCM) in Argentina is fragmented and has historically been based on estimates, supported only by a series of a few reported cases. Considering the lack of global information, a national multicentric study in order to carry out a more comprehensive analysis was warranted. We present a data analysis including demographic and clinical aspects of a historical series of 466 cases recorded over 10 years (2012–2021). Patients were aged from 1 to 89 years. The general male: female (M:F) ratio was 9.5:1 with significant variation according to the age group. Interestingly, the age range 21–30 shows an M:F ratio of 2:1. Most of the cases (86%) were registered in northeast Argentina (NEA), showing hyperendemic areas in Chaco province with more than 2 cases per 10,000 inhabitants. The chronic clinical form occurred in 85.6% of cases and the acute/subacute form occurred in 14.4% of cases, but most of these juvenile type cases occurred in northwestern Argentina (NWA). In NEA, the incidence of the chronic form was 90.6%; in NWA, the acute/subacute form exceeded 37%. Diagnosis by microscopy showed 96% positivity but antibody detection displays 17% of false negatives. Tuberculosis was the most frequent comorbidity, but a diverse spectrum of bacterial, fungal, viral, parasitic, and other non-infectious comorbidities was recorded. This national multicenter registry was launched in order to better understand the current status of PCM in Argentina and shows the two endemic zones with a highly diverse epidemiology.

## 1. Introduction

Paracoccidioidomycosis (PCM) is one of the most important endemic deep mycoses restricted to Latin American countries with the highest incidence in South America. Despite the fact that PCM was first described more than a century ago, its real incidence, prevalence, and regional epidemiological features remain undervalued. Its uneven distribution as well as it not being a notifiable disease, with the exception of Brazil, contribute to this lack of knowledge [1,2].

The information on PCM in Argentina is neither current nor real. Historically, only fragmented data are known, so the epidemiological characteristics of this systemic mycosis known until now are based on estimates, supported only by a series of scarce reported cases. The few national surveys about systemic mycoses show PCM as the most important endemic mycosis in the country [3].

Argentina comprises most of the southern endemic zone of PCM up to latitude 34° S, including two recognized endemic zones. The more extensive area located in northeast Argentina (NEA), bordering Brazil, Paraguay, and Uruguay, includes seven provinces or part of them: Chaco, Corrientes, Formosa, Misiones, Santiago del Estero, Santa Fe, and Entre Ríos. In northwest Argentina (NWA), the subtropical climate zones of Jujuy and Salta provinces represent the second endemic area in terms of extension and incidence [4].

In recent years, certain epidemiological changes in both endemic areas have been observed [4,5,6]. Considering these situations and the lack of global information at the country level, a national multicentric study of PCM in order to carry out a more comprehensive analysis was warranted. In order to establish the current status of PCM in Argentina, we present a data analysis including demographic, epidemiological, and clinical aspects in a historical series of the last 10 years.

## 2. Materials and Methods

Coordinated by the Mycology Department of Instituto de Medicina Regional, Universidad Nacional del Nordeste (Corrientes, Argentina), a national ambispective cohort multicentric study was conducted from January 2012 to December 2021 including sociodemographic, clinical, and laboratory diagnosis data on proven PCM cases. Only cases with a first diagnosis were included; recurrences and those who attended for medical control or treatment were not considered. The study was approved by the Ethics and Research Committee of the Instituto de Medicina Regional, Universidad Nacional del Nordeste, Argentina (Renis CE000326).

According to the European Organization for Research and Treatment of Cancer and the Mycoses Study Group Education and Research Consortium (EORTC/MSGERC) consensus, a proven PCM case was considered as an acute/subacute or chronic form of the disease diagnosed on epidemiological and clinical history and confirmed by direct microscopy or histopathology showing the distinctive multi-budding yeast cell consistent with *Paracoccidioides* spp. and/or recovery of the fungus by culture [7]. The global guideline for the diagnosis and management of the endemic mycoses also includes the considerations issued by the EORTC/MSGERC consensus and/or a positive antibody detection using the Ouchterlony immunodiffusion (ID) assay [8]. In our study, the ID method was performed using the *Paracoccidioides brasiliensis* B339 antigen produced by the reference center Instituto Nacional de Enfermedades Infecciosas “Dr. Carlos G. Malbrán” for the national mycology network.

The clinical forms of the disease were defined as the acute/subacute (“juvenile”) and the chronic (“adult”) form. The two major which are distinguished based on clinical aspects, the immunological response, and demographic characteristics [9,10,11].

Data collected from the medical records included age, sex, relationship with rural environment, risk factors (smoking and alcohol abuse), tissue/organ involvement, clinical manifestation, comorbidity, and the diagnosis method used. From each patient and/or the responsible family member, information about migratory history was also obtained including place of origin (hometown), municipalities they have lived or worked in, and previous occupations. For this analysis, patients who were diagnosed outside the endemic area were included in the endemic area where they were born, lived, or worked. Al data were recorded on standardized form and entered into a database in order to consolidate the information.

Georeferencing of all PCM clinical cases was carried out. Images were created with Adobe Illustrator CC2022, version 26.4.1^®^. Prevalence rates were calculated considering the number of inhabitants estimated on the records of the National Census of Population, Households and Housing in Argentina 2022 survey [12].

For the statistical analysis, associations were evaluated by a chi-squared (χ2) test, Pearson test, and Fisher’s exact test corrected. A significance level of 5% was considered statistically significant (*p* < 0.05). The odds ratio (OR) was applied to assess the chances of presenting a certain clinical manifestation depending on whether the patient suffered from an acute/subacute or a chronic clinical form. INFOSTAT statistical software was used to perform the data analysis [13].

## 3. Results

We analyzed data from 466 patients with proven PCM diagnosed in Argentina over a 10 year period, from January 2012 to December 2021, reported by 25 public and private hospitals or referral centers included in both endemic areas and outside of them, where the patient attended for diagnosis. The NEA area registered 384 cases and the NWA area registered 82 cases. Figure 1 shows the temporal distribution of PCM cases during this survey recording an average of 46.6 cases/year.

Figure 2 shows the geographical distribution of PCM cases according to the department of origin and number of cases per 10,000 inhabitants. The National Census of Population in Argentina 2022 survey records 1,635,380 inhabitants in the departments included in the NWA endemic area. In contrast, the accumulated population of the districts included in the NEA endemic zone reached 4,758,317 inhabitants.

Sex distribution stratified by age ranges of all patients including male: female (M:F) ratio is shown in Table 1.

The most prevalent clinical form in Argentina (85.6%) was the chronic form of the disease, with an average age of 55 and a high M:F ratio (13.8:1). Acute/subacute forms were observed in 14.4%, with an average age of 21 and a higher proportion of female cases (M:F ratio = 3.1) (Table 2).

A dissimilar prevalence of clinical forms was detected in both endemic areas. The distribution of clinical forms according to sex and the ratio for chronic: acute/subacute form for both areas is presented in Table 3.

Data about exposition to rural environment and alcohol consumption or smoking as predisposing habits are detailed in Table 2.

Organ involvement related to clinical forms of PCM is shown in Table 4. Most of the patients had more than one affected tissue. Disseminated forms occurred in 97% of the total cases and similar percentages are observed when the chronic and acute/subacute forms are analyzed separately (Table 5).

General symptoms such as weight loss were recorded in 65% of the patients, including all acute/subacute cases. Statistically significant differences were observed analyzing clinical manifestations and clinical forms of PCM. Localized or generalized adenomegaly (75.4%) and hepato-splenomegaly (32.3%) characterized most of the acute/subacute forms; in contrast, respiratory symptoms (51.2%), oropharyngeal lesions (50%), and cutaneous or mucocutaneous lesions (74%) were described in most of the chronic cases. In addition, the odds ratios showed that a chronic patient has more chances of having adrenal or central nervous system (CNS) involvement than a patient with an acute/subacute form. The main clinical manifestations are summarized in Table 5.

Chest radiographs were obtained in 20% of the patients with respiratory symptoms, which showed a general interstitial-alveolar pattern. Although some were nonspecific, most cases were bilateral, perihilar infiltrates mainly located in the middle third of the lungs, resulting in the typical “butterfly wing” pattern of PCM. Lung lesions as nodules were observed in seven patients: five chronic cases with multiple pulmonary nodules and two acute/subacute cases showing a unilateral nodule.

Except for seven acute/subacute cases, mucocutaneous lesions occurred almost exclusively in the chronic form (52.2%). Oropharyngeal manifestations predominated in the chronic form and lesions were also observed in infrequent areas: one perianal and the other on the penis (Table 5).

Polymorphic cutaneous manifestations including papules, papulonodular, ulcer-crusted, molluscoid, and scrofula-like lesions were recorded in 83 patients, 73.5% of whom were adults, and more frequently in areas of the head (45.1%) and trunk (32.9%). Lesions in facial skin, lips, nose, neck, chest, back, and abdomen were also registered (Table 5).

CNS manifestations were detected in 19 patients and 2 acute/subacute cases exhibited cerebellar abscesses. In chronic patients, the involvement of the CNS was manifested mostly (15/17) as cerebral nodules and in some (2/17), a cerebellar abscess was observed.

Eight cases showed osteoarticular lesions; six of them were in acute/subacute forms. Osteolytic lesions in long and cranial bones were evident in the infant–juvenile cases.

**Comorbidities.** Associated pathologies with PCM were observed in 47 patients. Tuberculosis (TB) was the most frequent, but a diverse spectrum of bacterial, fungal, viral, parasitic, and other non-infectious comorbidities was recorded (Table 2).

**Laboratory diagnosis.** Using microscopy and conventional culture, PCM was proved in 391/466 cases. In most cases (376/391) with 96% positivity, the diagnosis was made by microscopy, either by direct examination or histopathology.

In 75/466 (16%) cases, diagnosis was confirmed via antibody detection using ID associated to clinical and epidemiological data. As shown in Table 6, ID was reactive in 309/358 (86%) but 17% of the microbiological proven PCM patients were false negative.

In contrast to microcopy and serology, conventional culture was the less sensitive method for diagnosis, considering *Paracoccidioides* was isolated in only 24% of the 391 cultured clinical samples.

The performance of the ID assay in the microbiologically proven juvenile and chronic PCM form patients is shown in Table 7. A statistically significant difference *(p* < 0.05) depending on whether the patient is chronic or presents a juvenile form was observed.

## 4. Discussion

The impact of PCM on public health is not completely understood because of the lack of accurate information (precise data) on PCM incidence since it is not a compulsory notification disease in most Latin American countries; Argentina is not an exception to this reality. For this reason, it was decided to create this national multicenter study on PCM in order to have retrospective as well as prospective and updated information at the country level and particular level of each region, which allows the situation to be exposed and to work on the health policies that are required.

During the last 10 years, Argentina has shown a slight but constant increase in the number of cases/years. In 2020, the frequency of cases decreased to extremely low records as a consequence of the COVID-19 pandemic. While health systems tried to control the crisis, other pathologies were relegated. On the other hand, the population locked themselves in and fear prevented them from consulting or attending medical check-ups, even for those who already had a diagnosis. As Figure 1 shows, in 2021 the frequency of cases did not reach the level of previous years. If the COVID-19 pandemic had not occurred, surely this series would show more than 500 cases. Unlike other fungal respiratory infections, no relationship between COVID-19 and PCM was documented [14,15].

Due to the higher number of registered cases (82%) and its territorial extension, our study demonstrates that the NEA endemic area is the Argentine zone with the highest incidence of PCM, but also because it includes the departments with the highest frequency of cases per 10,000 inhabitants, as can be seen on the cases’ georeferencing map (Figure 2). Chaco was historically the province with the highest PCM incidence in Argentina [16,17,18]. This study confirms that Chaco is a hyperendemic area of PCM in NEA. In the NWA area, the subtropical jungle region of Las Yungas in the province of Salta registers the highest number of cases per 10,000 inhabitants, but this is always lower than what is observed in Chaco.

Demographic features may influence infection with *Paracoccidioides* and the clinical course. PCM was historically related to living and/or working in rural areas. It is noteworthy that in our multicentric study, only 47.23% of the cases referred to exposure in a rural environment, while the rest were registered in urban or peri-urban backgrounds. In this regard, changes in the geographic and demographic patterns of the population with PCM have been reported in recent years, including the occurrence of PCM in urban areas and on the periphery of urban centers (overlapping with rural zones) [19,20,21,22,23]. In Argentina, urban and peri-urban cases were also reported in a series of infant–juvenile PCM [6]. On the other hand, it would be important to analyze the term “urban” to define the demographic characteristics of a clinical case. In Latin America, the criteria for determining an area as urban differ considerably between countries; therefore, many so-called urban areas are localities with a very close rural environment with different endowment of infrastructure, population density, and surface area. In Argentina, an area is considered urban when it has more than 2000 inhabitants, urbanization infrastructure, and basic services such as drinking water and electricity networks.

Since estrogens inhibit mycelium-to-yeast transformation, PCM was described mostly in men [24]. Furthermore, the chronic clinical form is prevalent in men as a consequence of the higher exposure to the fungus found in the soil, due to rural work carried out mainly by men [22,25,26]. The general M:F ratio obtained in Argentina was 9.5:1, lower than reports from other countries [9,27,28], but this proportion showed a marked variation according to the age group (Table 1). Although there are many reports on PCM from other countries, not all of them stratify the M:F ratio by age range; therefore, it is difficult to compare since age influences the appearance of the disease in females. Currently, women are more involved both in management and in rural work. This higher exposure to the inoculum may be one of the reasons for this lower M:F ratio observed in our study.

Approximately 80% of the cases occurred in patients older than 41 years, with the highest prevalence between 51–60 years, but it is striking that the highest M:F ratio was observed between 61–70 years, when women have the same susceptibility as men. Since the infection is considered to be acquired mainly in the first two decades of life [9], these findings show how PCM can manifest (appearance of clinical manifestations or disease development) many decades after infection. Since all the cases analyzed are newly diagnosed chronic cases, some may have acquired the infection up to four decades earlier.

In children, it is well known that the M:F relationship is low, which increases from adolescence due to the protective effect of estrogen [29]. Interestingly, the opposite occurs in our study. Analyzing the age range of 21–30 years, when hormonal activity is higher, the ratio was 2:1. PCM patients included in this age range live in rural areas where agriculture is carried out. On the other hand, no woman in this group reported being a smoker or an alcohol consumer. Possible explanations are that women are currently more involved in rural work, but also that the use of agrochemicals in these regions is not well controlled/regulated and many of them have endocrine disrupting properties affecting estrogen signaling. These compounds mimic or block the actions of endogenous hormones, alter synthesis or secretion, or interfere with downstream actions that would typically occur when a hormone binds its receptor [30]. In this case, the protective action of estrogen would be inhibited and, for this reason, the M:F ratio is low. Further studies are required to analyze and confirm this effect.

The effect of living in a rural area is also reflected in the prevalence of older adults and the elderly registered in this study, which reaches up to 89 years old. Most of these cases were diagnosed late or the patient came with advanced disease. The low clinical suspicion, the centralization of the diagnosis, the lack of generation of more precise and point-of-care testing, which together cause a low testing rate, were exposed as the current problematic weaknesses of PCM in the Americas [31].

Tobacco and alcohol abuse associated with PCM in chronic patients has been well established [2,32]. The risk of developing this mycosis is 14 times more superior among smokers than among non-smokers [10,33]. In our survey, these predisposing factors were registered in 28.3% with a high relationship for men with the chronic clinical form. We are sure that these are not the real values, and it is an underreporting as a consequence of the fact that patients do not want to report their true addictions. Many of them do not attend accompanied and, therefore, it is not possible to inquire about their true situation with the family.

Rates between 3–25% have been reported for the juvenile-type PCM in Latin America [9,19,23,34]. In Argentina, the acute/subacute (juvenile) form registers 14.2%, but most of these cases occur in NWA. In NEA, the incidence of the chronic form is overwhelming (90.6%); in NWA, the acute form exceeds 37%. This particularity of this endemic area was historically observed, but poorly documented. This is the first publication analyzing a series of cases over a long period of time confirming these epidemiological features in NWA.

For unifocal or associated respiratory symptoms with an abnormal chest radiograph, lymph nodes were compromised, but also mucocutaneous lesions were more related to the chronic form following the standards reported [25,28,35,36]. Isolated pulmonary lesions require a differential diagnosis with TB. The association of PCM with TB is frequent; moreover, PCM is commonly misdiagnosed as TB [35,36]. When only pulmonary abnormalities are present, epidemiological investigation and imaging findings are crucial, but laboratory investigations are required to arrive at an accurate diagnosis.

The expected range of PCM/TB association is 5–10% and can also be higher [9,21,35]. Our value is lower than the other series reported in Brazil [22,28,37]. Although this comorbidity was not high in our survey, we believe that some confusion often results not only in the delayed diagnosis and treatment of PCM, but also in an underdiagnosis of the associations. The differential diagnosis of the PCM/TB combination can be difficult on the basis of clinical and radiological data. It is necessary to carry out an exhaustive bacteriological and mycological investigation prior to instituting a specific therapeutic regimen.

Mucocutaneous lesions led to the diagnosis in more than half of the cases (64%). Ty-pical lesions of the oral mucosa, ulcers, or ulcer-vegetative form with hemorrhagic dots (moriform stomatitis) are reported in between 60% and 75% of chronic forms of PCM [25]. Manifestations in the nasal cavities, oral cavity, oropharynx, hypopharynx, and larynx constitute the upper aerodigestive tract (UADT) involvement and are of high prevalence [38]. These lesions provoke a high degree of PCM suspicion; the fungal load observed is generally high and sampling is minimally invasive. Training and experience in observation by clinicians, and especially otolaryngologists, becomes an important tool for differential diagnosis in endemic areas or in patients with a history of having lived or worked in this environment.

Although cutaneous manifestations are less frequent, they also facilitate diagnosis due to their easy access for direct microscopy examination and biopsy, when PCM is considered. Many patients in this series had single skin lesions without mucosal manifestations. The polymorphism of these isolated lesions, ranging from disseminated papules, crusted lesions, other granulomatous lesions, and even scrofula-like lesions, delayed diagnosis because they were not “thought” to be a PCM manifestation, but rather another differential diagnosis.

More than 21% of the acute/subacute form cases were children aged 1–10 years. Findings in patients with the juvenile clinical form were comparable to other series [19,23,25,28]. Lymph node compromise was present in both clinical forms, but adenomegaly was more frequently (77.3%) observed in children and young people. The prompt lymphatic-hematogenous dissemination of the fungus, with signs and symptoms of severe disease, make young patients vulnerable, who in many cases find a late diagnosis. Even within the endemic area, the differential diagnosis is usually a challenge, either due to the high incidence of other regional pathologies with similar signs or symptoms, or the rapid general worsening that can confuse the clinical diagnosis. This problem was observed in PCM outbreaks in NEA and in other reports from the endemic area of Brazil [6,19]. Unlike the adult form, mucosal involvement typically occurs in rare cases and skin lesions do not guide the PCM diagnosis due to the diversity of types of lesions. In our series of cases, the skin manifestations were presented as papules, fistulized and non-fistulized abscesses, nodules, tumors, and ulcers in different anatomical sites. In most cases, the diagnosis was obtained from the puncture or histopathology of lymph nodes.

In this study, all infant–juvenile cases manifested weight loss and most lymph node enlargement. Interestingly, children from NWA showed liver and spleen compromise and digestive symptoms while skin, mucocutaneous, and bone involvement were mostly observed in children from the NEA area. Further analysis and studies are required to determine the reason for these differences between both endemic regions. One of the many questions to be resolved is to know the cryptic species of *Paracoccidioides* circulating in both endemic regions and if they have any relationship with the clinical form. Although some advances were obtained, the knowledge that is currently available is scarce [39].

The frequency of PCM co-infection with HIV has historically been low. In Brazil, the PCM/HIV-infected rate was estimated at 0.33 cases per million people based on 200 cases [40]. In our study, 14 cases (3.5%) were registered, more related to the chronic form, and it was consistent with the prevalence rates reported in other series [21,22]. PCM/HIV presents most frequently as a disseminated disease with differences in respect to the classic form’s presentation. Relapses and higher mortality rates are reported in PCM/HIV patients as a consequence of the low clinical suspicion of this coinfection, late diagnosis, and uncertainties about its management [21,41]. In our series, PCM/HIV coinfections with cryptococcosis, histoplasmosis, HBV, and Chagas disease were registered. The different pattern of symptoms, characterized by a more acute and severe clinical picture with overlapping of symptoms and manifestations, makes the clinical diagnosis of these coinfections difficult and, therefore, the prognosis is worse. PCM is also related to other immunosuppressive diseases. In our survey, associations to neoplasia and solid organ transplantation (SOT) were registered in low frequency. Laboratory diagnosis in immunosuppressed patients can be challenging. Antibody detection generally has low sensitivity and antigen detection tests are not available; therefore, this association may be underdiagnosed [41].

Chagas disease (1.25%), cutaneous leishmaniasis (0.75%), and strongyloidiasis (0.25%) association were detected in chronic patients. These diseases, as well as TB and PCM, all share the same predisposing conditions, such as low socioeconomic status, malnutrition, and exposure to rural and suburban areas, which is consistent with similar observations in other reports [22,28].

Since new and more sensitive tools such as the detection of *Paracoccidioides* antigens have not been developed nor specific PCR methods are available in routine laboratories, the gold standard method for diagnosis remains as microscopic visualization of typical multi-budding yeasts by histopathology or direct mycological examination (wet mounts and stains) and isolation of *Paracoccidioides* spp. in culture [31,32,38]. In our study, microscopy (direct or histopathology) allowed the diagnosis with higher (96%) sensitivity than other studies that show an average of 48–75% [36]. This sensitivity also depends on the experience of the operator and the quality of the clinical sample. In contrast, the low sensitivity of the culture was also observed in our results (24%). Being a fastidious and slow-growing fungus, this limits its use for early diagnosis. On the other hand, many Latin American laboratories do not perform culture, but only direct examination, decreasing the sensitivity of the conventional diagnosis [31].

The detection of *Paracoccidioides* spp. antibodies is one of the most frequently used diagnostic tools. Even more, this method may be the only one, when access to the clinical sample is difficult or we do not have a positive microscopy. Unfortunately, there is not a commercial test available to detect the currently known *Paracoccidioides* species. Antibodies and antigens are prepared in-house from the reference strain Pb B339, but not from autochthonous strains. Antigens obtained from Pb B339, rich in the immunodominant antigen glycoprotein 43 KDa (gp43), have been reported to be accurate in the diagnosis of *P. brasiliensis*, but have low sensitivity in *P. lutzii* disease cases [9,38]. Studies carried out in different zones of the Brazilian endemic area demonstrated a low level of concurrence using only the Pb B339 antigen. Sera from patients from the Center-West Region of Brazil could not precipitate the reference antigen [42]. In our study, 17% of the ID tests that were performed were false negative in the microbiologically proven PCM patients. In the acute/subacute cases, this percentage increased to 30%, a very high figure when only this test is available to make a diagnosis. We used an antigen from a non-autochthonous strain; the different antigenic composition is probably the cause of these results.

It is estimated that around 10 million people are infected in South America with incidence rates from 1–3 cases/100.000 inhabitants/year to 9–15 cases/100.000 inhabitants/year in hyperendemic areas [9,10,43]. PCM is frequently misdiagnosed and can be associated with or mimic infectious and non-infectious disorders, such as TB, cancer, sarcoidosis, and even other systemic mycoses such as histoplasmosis and cryptococcosis, as we have found in this series. The limited tests available for diagnosis require a high index of suspicion. Rapid recognition of this mycosis is crucial in the juvenile form to prevent mortality and in the chronic form to avoid, if possible, serious sequelae such as pulmonary fibrosis and tracheal stenosis. Pitfalls in its recognition and management are common [44]. Endemic mycoses are not usually first considered during the initial or subsequent evaluation and the low degree of awareness and education among health professionals and public health authorities was exposed at the last International Meeting on Endemic Mycoses of the Americas [31]. Several explanations contribute to the fact that this endemic systemic mycosis is among those considered as neglected, but that status has not yet been declared. Neglected tropical diseases (NTD) affect Latin American countries and PCM is one of the most frequent systemic mycoses [43,45]. In 2022, *Paracoccidioides* was included in the World Health Organization’s (WHO) list of fungal priority pathogens categorized within the Medium Priority Group [46]. PCM fulfils WHO’s NTD criteria and the Latin American community would benefit from such a classification. The Latin American scientific community argues that it should be explicitly recognized [31,47].

## 5. Conclusions

The impacts of PCM are an important cause of mortality in infants and the young population, and they lead to disabling conditions in adults that reduce productivity and quality of life [10,48]. With the exception of Brazil, in Argentina and most Latin American countries, PCM is not a notifiable disease; therefore, there is a lack of surveillance and health policies.

This national multicenter registry was launched in order to better understand the current status of PCM in Argentina. Its incidence has shown a slow but constant increase in the last 10 years, but a drastic reduction was observed in 2020–2021, as a consequence of the COVID-19 pandemic.

In contrast to other countries, a low M:F ratio (9.5:1) is observed with significant variation according to the age group. Despite the protective effect of estrogens, this ratio drops to 2:1 in the age range from 20 to 30 years. Our next step will be to carry out further studies to analyze whether the possible causes discussed in this paper generate this higher incidence of PCM in sexually active women.

In Argentina, there is an extensive endemic zone with the largest population in the NEA part of the country and a smaller one in the NWA part of the country. This study showed both endemic areas with diverse epidemiological features.

Most of the cases (86%) were registered in the NEA area, where Chaco province shows hyperendemic areas with more than 2 cases per 10,000 inhabitants.

The acute/subacute clinical form occurs in Argentina in a higher percentage (14.4%) than in other Latin American endemic areas, but most occur in NWA. In the NEA, the incidence of the chronic form was 90.6%; in NWA, the acute form exceeded 37%.

The infant–juvenile forms comprising the NWA area show different clinical manifestations to those cases from the NEA area. Children from NWA showed liver and spleen compromise and digestive symptoms while skin, mucocutaneous, and bone involvement were mostly observed in children from the NEA area.

Detection of antibodies is one of the main tools for PCM diagnosis but it shows 17% of cases to be false negatives, thus necessitating the improvement in this method when the conventional method is not accessible.

PCM is one of the most important endemic deep mycoses in Argentina. This series of cases reflects the state of the art during the last 10 years, thanks to case reports from all the centers in Argentina that diagnosed patients with this endemic systemic mycosis.

This national cohort multicentric study will continue to provide accurate and up-to-date information to assess the dynamics of PCM in Argentina.

## Figures and Tables

**Figure 1 jof-09-00482-f001:**
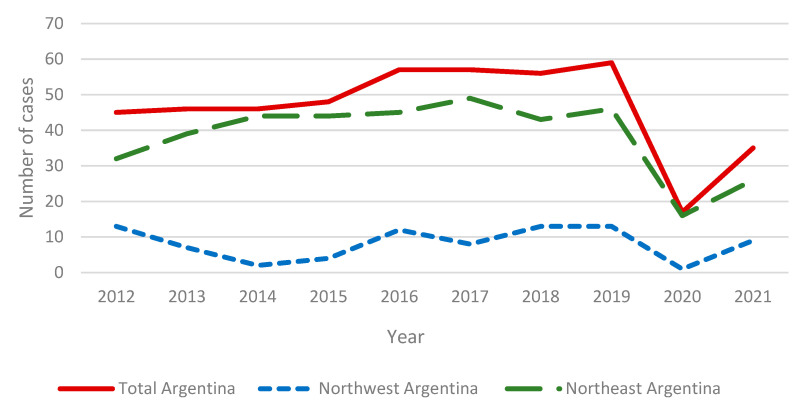
Temporal distribution of PCM cases in the period 2012–2021.

**Figure 2 jof-09-00482-f002:**
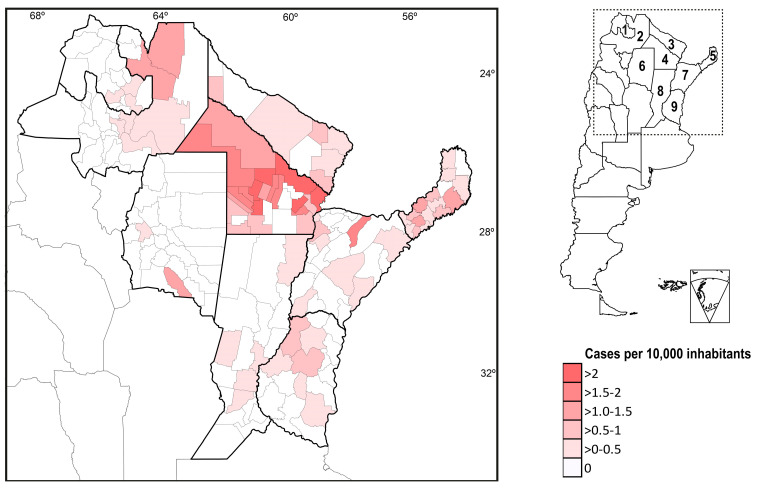
Geographical distribution of PCM cases assigned by provincial departments. Provinces: (1) Jujuy and (2) Salta are included in the northwest endemic region (NWA). Provinces: (3) Formosa, (4) Chaco, (5) Misiones, (6) Santiago del Estero, (7) Corrientes, (8) Santa Fe, and (9) Entre Ríos integrate the northeast endemic zone (NEA).

**Table 1 jof-09-00482-t001:** Sex/age distribution of 466 PCM patients recorded in Argentina.

Age Range	Male		Female		Total		M:F Ratio
	n		n		n		
0–10	14	3.3%	3	7.0%	17	3.6%	4.7:1
11–20	14	3.3%	4	9.3%	18	3.9%	3.5:1
21–30	13	3.1%	6	14.0%	19	4.1%	2.1:1
31–40	43	10.2%	4	9.3%	47	10.1%	10.7:1
41–50	91	21.5%	7	16.3%	98	21.0%	13:1
51–60	135	31.9%	13	30.2%	148	31.8%	10.4:1
61–70	93	22.0%	4	9.3%	97	20.8%	23.2:1
71–80	17	4.0%	2	4.7%	19	4.1%	8.5:1
81–89	3	0.7%	0	0%	3	0.6%	3:0
	423		43		466		9.5:1

**Table 2 jof-09-00482-t002:** Sex distribution analysis considering clinical form, exposure/habits, and comorbidities of 466 clinical cases recorded.

	Male		Female		Total	
	n	%	n	%	n	%
**Clinical form**						
Acute/subacute	50	75.8%	16	24.2%	66	14.2%
Chronic	373	93.3%	27	6.8%	400	85.8%
**Rural exposure/occupation**	218	96%	8	4%	226	
**Habits**						
Smoker	80	98.7%	1	1.3%	81	
Alcohol abuse	32	100%	0	0%	32	
**Comorbidities**						
TB	12		0		12	
TB + Chagas disease	1		0		1	
Chagas disease	3		0		3	
HIV	7		3		10	
HIV + cryptococcosis	1		0		1	
HIV + histoplasmosis	1		0		1	
HIV + HBV	0		1		1	
HIV + Chagas disease	1		0		1	
Leishmaniasis	3		0		3	
Carcinoma	1		0		1	
Strongyloidiasis	1		0		1	
Cryptococcosis	2		0		2	
Aspergillosis	1		0		1	
Syphilis	1		0		1	
HBV	1		1		2	
Aortitis Leriche syndrome	1		0		1	
SOT	0		1		1	
DBT	2		1		3	
Metabolic syndrome	0		1		1	

TB: tuberculosis; HIV: human immunodeficiency virus; HBV: hepatitis B virus; SOT: solid organ transplantation; DBT: diabetes.

**Table 3 jof-09-00482-t003:** Clinical forms and sex distribution in northeast (NEA) and northwest (NWA) Argentinian areas.

	Northeast Argentina							Northwest Argentina						
Clinical Form	Male		Female		Total		Ratio C:A/S ^1^	Male		Female		Total		Ratio C:A/S ^1^
Chronic (n = 400)	328	92%	20	77%	348	90.6%		45	69%	7	41%	52	63%	
Acute/subacute (n = 66)	30	8%	6	23%	36	9.4%		20	31%	10	59%	30	37%	
Total	358		26		384		9.7:1	65		17		82		1.7:1

^1^ Ratio chronic: acute/subacute form.

**Table 4 jof-09-00482-t004:** Frequency of organ or tissue involved according to clinical form registered in the studied cohort.

Organ/Tissue Involvement	Total Cases	Clinical Form		OR—*p* Value (*p*)
	n	Acute/Subacute	Chronic	
Lung (X-ray evaluated)	211	10 (4.7%)	201 (95.3%)	OR C/A = 5.66, *p* < 0.05
Palate	54	2 (3.7%)	52 (96.3%)	OR C/A = 4.36, *p* < 0.05
Tongue	22	0	22 (100%)	OR C/A = 3.78, *p* < 0.05
Cheek mucosa	87	2 (2.3%)	85 (97.7%)	OR C/A = 8.63, *p* < 0.05
Lips	30	0	30 (100%)	OR C/A = 5.27, *p* < 0.05
Larynx	27	0	27 (100%)	OR C/A = 4.71, *p* < 0.05
Pharynx	3	2 (66.7%)	1 (33.3%)	OR A/C = 12,47, *p* < 0.05
Amygdala	3	0	3 (100%)	OR C/A = 0.66, *p* > 0.05
Epiglottis	4	1 (25%)	3 (75%)	OR C/A = 0.49, *p* > 0.05
Nasal mucosa	14	0	14 (100%)	OR C/A = 2.53, *p* > 0.05
Suprarenal gland	12	1 (8.3%)	11 (91.7%)	OR C/A = 1.84, *p* > 0.05
Lymph nodes	75	49 (65.3%)	26 (34.7%)	OR A/C = 41.46, *p* < 0.05
Skin	61	17 (27.9%)	44 (72.1%)	OR A/C = 2.81, *p* < 0.05
Liver	27	24 (88.9%)	3 (11.1%)	OR A/C = 75.66, *p* < 0.05
Spleen	23	21 (91.3%)	2 (8.7%)	OR A/C = 92.87, *p* < 0.05
Bone/joint	8	5 (62.5%)	3 (37.5%)	OR A/C = 10.85, *p* < 0.05
Cerebrum	14	0	14 (100%)	OR C/A = 2.53, *p* > 0.05
Cerebellum	5	3 (60%)	2 (40%)	OR A/C = 9.48, *p* < 0.05

OR: odds ratio.

**Table 5 jof-09-00482-t005:** Main clinical manifestations according to clinical form of 466 patients recorded during 2012–2021.

Clinical Manifestations	Total Cases	Clinical Form		OR—*p* Value (*p*)
	n = 466	Acute/Subacute	Chronic	
Localized	12 (3%)	2 (3.3%)	10 (2.5%)	
Disseminated	454 (97%)	64 (96.7%)	390 (97.5%)	
Weight loss	301	66 (21.9%)	235 (78.1%)	OR A/C = 52.65, *p* < 0.05
Respiratory symptoms	206	8 (3.9%)	198 (96.1%)	OR C/A = 7.11, *p* < 0.05
Oropharyngeal lesions	197	7 (3.6%)	190 (96.4%)	OR C/A = 7.63, *p* < 0.05
Periodontitis	62	0	62 (100%)	OR C/A = 11.92, *p* < 0.05
Laryngeal	23	0	23 (100%)	OR C/A = 3.97, *p* > 0.05
Nasal mucosa ulceration	14	0	14 (100%)	OR C/A = 2.36, *p* > 0.05
Perianal ulceration	1	0	1 (100%)	
Genitals ulceration	1	0	1 (100%)	
Cutaneous lesions				OR C/A = 2.89, *p* < 0.05
Face (facial skin, lips)	26	3 (11.5%)	23 (88.5%)	
Nose	6	0	6 (100%)	
Eyelid	2	1 (50%)	1 (50%)	
Scalp	3	1 (33.3%)	2 (66.7%)	
Ear	1	0	1 (100%)	
Neck	9	8 (88.9%)	1 (11.1%)	
Arms	2	1 (50%)	1 (50%)	
Hand/finger	2	0	2 (100%)	
Trunk (chest, back, and abdomen)	27	8 (29.6%)	19 (70.4%)	
Leg	2	0	2 (100%)	
Foot	3	0	3 (100%)	
Localized adenomegaly	65	40 (61.5%)	25 (38.5%)	OR A/C = 23.08, *p* < 0.05
Generalized adenomegaly	13	11 (84.6%)	2 (15.4%)	OR A/C = 39.80, *p* < 0.05
Adrenal disorders	12	1 (8.3%)	11 (91.7%)	OR C/A = 1.84, *p* > 0.05
Hepatomegaly	24	21 (87.5%)	3 (12.5%)	OR A/C = 61.76, *p* < 0.05
Splenomegaly	23	21 (91.3%)	2 (8.7%)	OR A/C = 61.76, *p* < 0.05
Osteoarticular	8	6 (75%)	2 (25%)	OR A/C = 19.90, *p* < 0.05
Central nervous system	19	2 (10.5%)	17 (89.5%)	OR C/A = 1.42, *p* > 0.05

OR: odds ratio.

**Table 6 jof-09-00482-t006:** Applied methods for the diagnosis of paracoccidioidomycosis.

Method	Total	Positive		Negative	
		n	%	n	%
Microscopy	391	376	96%	15	4%
Conventional culture	391	95	24%	296	76%
Immunodiffusion	358	309	86%	49	14%
ID (+ microscopy + culture)	283	234	83%	49	17%
Only ID	75	75	100%	0	

**Table 7 jof-09-00482-t007:** Immunodiffusion (ID) performance in microbiological proven juvenile and chronic forms.

Clinical Form	Total ID Performed	Positive		Negative	
		n	%	n	%
Acute/subacute	54	38	70%	16	30%
Chronic	304	271	89%	33	11%

Odds ratio chronic/acute subacute = 3.46—*p* < 0.05.

## Data Availability

Data presented in this study are available upon request from the corresponding author.

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
