# Peer review of "Clinical and Demographic Features of Paracoccidioidomycosis in Argentina: A Multicenter Study Analysis of 466 Cases"

_jof, 2023, doi:10.3390/jof9040482_

Round 1

Reviewer 1 Report

This interesting study addresses Paracoccidioidomycosis's clinical and demographic features in Argentina. In general, the study was well conducted, and the manuscript is appropriately written, for which I compliment the authors. Below, I present some suggestions that may be useful for further enhancement of the manuscript's quality.

1. Title. The title is informative.

2. Abstract. The abstract lacks the study design. The results presentation and conclusion are somehow imprecise, as they do not state what are the main differences between the two endemic zones' epidemiology.

3. Keywords are appropriate.

4. Introduction reads well.

5. Methods.

5.a I'm unsure if this can be classified as a cohort study since the authors do not present patient outcomes. It seems to me this was originally a survey that was analyzed cross-sectionally.

5.b It is also unclear how patients were recruited since the disease is not identifiable in Argentina. I mean, did authors search for patients both in public and private services? Recruitment was apparently focused on hospitals, but what about the mild cases seeking care in outpatient clinics?

5.c Please replace "gender" with "sex". There are several kinds of gender but only two sexes.

5.d Despite PCM being a chronic disease, the word "prevalence" is not appropriate because this is a curable disease. As the authors are studying only the new cases of PCM, the term "incidence" is more appropriate.

5.e. The phrase "Possible associations between clinical forms and organ/tissue involvement but also clinical manifestations of the disease were evaluated" is ambiguous to me. Please rewrite.

5.f. Authors need not explain what the interpretation of a given odds ratio is. A typical reader of any scientific journal is not naive to that.

6. Results. The results section is quite interesting and Figure 2 is very ilustrative.

6.a Table 2 presentation is very unusual, in my opinion. Normally, in cross-sectional studies, we put the dependent variables (outcomes) in the column and the independent variables (exposure/risk factor) in the rows. Therefore, sex distribution, which is an absolutely independent variable, should not be presented in a column. Please rearrange the table.

6.b Regarding Table 4, please add % to the numbers presented. I did not understand why the authors chose to apply a statistical test to compare the clinical differences between acute and chronic forms of the disease. As far as I understood, the study was based on a whole population, not on a sample. Therefore, the differences observed are "real" and not "estimated", which turns inappropriate any statistics test application.

6.c The same applies to table 5.

6.d Regarding table 6, please show the positivity for each individual test, not combinations.

7. Discussion.

7.a The discussion is interesting but goes too long and could benefit from shorting. Please avoid excessive comparisons with other studies regarding the clinical presentation.

7.b PCM may not be notifiable in the whole South America, but it is notifiable in Brazil. Please acknowledge that.

7.c The discussion lacks a conclusion. Please harmonize with the abstract's renewed conclusion.

Author Response

Abstract. The abstract lacks the study design. The results presentation and conclusion are somehow imprecise, as they do not state what are the main differences between the two endemic zones' epidemiology.

Response 1: Thank you for your observation, you are right. It is very difficult to give more details in an abstract that cannot exceed 200 words. The abstract was revised. The main differences between the epidemiology of the two Argentine endemic areas were added.

 Methods.

5.a I'm unsure if this can be classified as a cohort study since the authors do not present patient outcomes. It seems to me this was originally a survey that was analyzed cross-sectionally.

Response: This study was classified as a cohort study based on the definition considering a cohort study a particular form of longitudinal study that samples a cohort (a group of people who share a defining characteristic), performing a cross-section at intervals through time. A type of panel study where the individuals in the panel share a common characteristic, in this case PCM.

Do you think we should change the term or delete it?

5.b It is also unclear how patients were recruited since the disease is not identifiable in Argentina. I mean, did authors search for patients both in public and private services? Recruitment was apparently focused on hospitals, but what about the mild cases seeking care in outpatient clinics?

Response: Patients were recruited from both public and private services, including referral centers. Some hospitals included in this study are private hospitals. There are few outpatient clinics in the endemic areas and, although the disease is not notifiable in Argentina, when a case (mild or not) appears, they are transferred to a hospital or referral center, because they do not have a mycology laboratory for an accurate diagnosis. On the other hand, in Argentina there is a national mycology network. This multicenter study is also supported by this network. In this way, all hospitals, health centers and referral centers report their cases to this national multicenter. Even so, we cannot ensure that all the patients included in this study represent 100%, but we understand that it is a representative sample.

5.c Please replace "gender" with "sex". There are several kinds of gender but only two sexes.

Response: Thanks for your comment, gender was replaced by sex

5.d Despite PCM being a chronic disease, the word "prevalence" is not appropriate because this is a curable disease. As the authors are studying only the new cases of PCM, the term "incidence" is more appropriate.

Response: The study was retrospective and prospective, for this reason the word prevalence was used. Do you think we should change the term prevalence in all cases?

5.e. The phrase "Possible associations between clinical forms and organ/tissue involvement but also clinical manifestations of the disease were evaluated" is ambiguous to me. Please rewrite.

Response: To clarify, the phrase was rewrited.

5.f. Authors need not explain what the interpretation of a given odds ratio is. A typical reader of any scientific journal is not naive to that.

Response: Interpretation was deleted.

  1. Results. The results section is quite interesting and Figure 2 is very ilustrative.

6.a Table 2 presentation is very unusual, in my opinion. Normally, in cross-sectional studies, we put the dependent variables (outcomes) in the column and the independent variables (exposure/risk factor) in the rows. Therefore, sex distribution, which is an absolutely independent variable, should not be presented in a column. Please rearrange the table.

Response: Thanks for your opinion. I agree with "normally". Considering that in PCM the differences between men and women are important to observe and highlight, for this reason the table was organized in this way. Otherwise, a table would be difficult to insert in the journal edition or would have to be split.

Since the other 2 reviewers considered the table to be correct, it was not changed.

6.b Regarding Table 4, please add % to the numbers presented. I did not understand why the authors chose to apply a statistical test to compare the clinical differences between acute and chronic forms of the disease. As far as I understood, the study was based on a whole population, not on a sample. Therefore, the differences observed are "real" and not "estimated", which turns inappropriate any statistics test application.

Response: % were added.

We consider it relevant to know which organ or tissue is most affected according to the clinical form. And really statistically significant differences were observed. In some cases, this could help us to define the clinical form of the patient, which between the ages of 30 and 40 usually raises doubts.

6.c The same applies to table 5.

Response: % were added.

  1. Discussion.

7.a The discussion is interesting but goes too long and could benefit from shorting. Please avoid excessive comparisons with other studies regarding the clinical presentation.

Response: You are right. The large amount of data and its richness motivated such a long discussion, but we did not want to leave data without discussion. Excessive comparisons with other studies regarding the clinical presentation were avoided.

7.b PCM may not be notifiable in the whole South America, but it is notifiable in Brazil. Please acknowledge that.

Response: Absolutely true. It was added.

7.c The discussion lacks a conclusion. Please harmonize with the abstract's renewed conclusion.

Response: Conclusion harmonized with the abstract was added.

Reviewer 2 Report

The authors review the clinical data regarding paracoccidioidomycosis in Argentina. The topic is likely needed to fill out demographic information regarding this disease. The conclusions are consistent with the data.

For the most part well written and comprehensive. Some of the wording is clumsy and hard to understand. The Discussion could be shortened.

Author Response

Thank you so much for your comments and suggestions.

I am attaching the new version of the manuscript including your and Reviewer 1 and 2 observations.

Reviewer 3 Report

The research article submitted by Giusiano et al. is a good information to fill an existing gap in relevance to epidemiology. Here I have some comments towards the improvement of the submitted draft:

1. Introduction is very weak, it requires more information regarding the causal organism and stats at global and national scale. Some recent examples/Clinical surveys/Health data (2021-2023) should be included to clear the current clinical status and the need of the study.

2. At line no. 89: Please provide the full forms of “ORTC/MSGERC”. Similarly, please check throughout the text for other short forms used.

3. Author’s have provided geographical distribution with a map. Its important to add latitudes and longitudes to the map, without this geographical information, this is incomplete.

4. Gender distribution of PCM patients is well discussed with suitable studies. Also, the decrease of the cases and its reason during 2020, as COVID impact.

5. Data can be shown in the form of graphs/charts which is self-explanatory. Currently, all the data is in table which is less effective to understand.

6. In the Table 1. (As very first table), Author’s need to describe “n” as total cases. It can be added at the bottom of the table.

7. Tables provided in the draft is very low quality. No identical pattern/formatting is followed. Please improve the quality of the tables as per journal guidelines.

8. Figure 1. Please label the x and y axis of the graph. NWA and NEA are not described in the legends, please provide full forms in the legend. 

Author Response

Thank you so much for your accurate comments and suggestions.

The requested corrections were made on the text, map, and tables.

Your observation about de tables format is correct. All the tables were designed according to the journal's guidelines, but apparently the system changed them. We also received the manuscript with the formats of all the tables changed. We have arranged all of them.

You requested to include in Introduction some recent examples/Clinical Surveys/Health Data (2021-2023) to clarify the current clinical status and need for the study. The lack of these data and the lack of knowledge about the current clinical status generated the need for this study. In Introduction, 3 previous works that prompted the realization of this larger study were referenced.

I am attaching the new version of the manuscript including your and Reviewer 1 and 2 observations.
